# Association between Rumen Microbiota and Marbling Score in Korean Native Beef Cattle

**DOI:** 10.3390/ani10040712

**Published:** 2020-04-19

**Authors:** Minseok Kim, Tansol Park, Jin Young Jeong, Youlchang Baek, Hyun-Jeong Lee

**Affiliations:** 1Animal Nutrition & Physiology Team, National Institute of Animal Science, Wanju 55365, Korea; mkim2276@gmail.com (M.K.); jeong73@korea.kr (J.Y.J.); chang4747@korea.kr (Y.B.); 2Department of Animal Science, College of Agriculture and Life Sciences, Chonnam National University, Gwangju 61186, Korea; 3Department of Animal Sciences, The Ohio State University, Columbus, OH 43210, USA; tansol1719@gmail.com; 4Dairy Science Division, National Institute of Animal Science, Cheonan 31000, Korea

**Keywords:** Hanwoo, beef cattle, marbling, rumen microbiota, lipid metabolism

## Abstract

**Simple Summary:**

The ruminal microbiome affects various metabolic processes associated with animal development; however, few studies have focused on its correlation with marbling. Results of the present study show differences in ruminal microbiomes among Hanwoo Korean beef cattle, which have low or high marbling scores. By elucidating the effect of the ruminal microbiome on the marbling of Hanwoo, differentially abundant microbial taxa, ruminal taxonomic drivers of lipid metabolism, and the correlation with meat quality indices, the present study provides insights into the potential effects of microbial factors on marbling in beef cattle.

**Abstract:**

This study demonstrated the potential effects of the rumen microbiota on the deposition of intramuscular fat, known as marbling. Previous studies on fatty acid metabolism in beef cattle have mostly focused on biohydrogenating rumen bacteria, whereas those on the overall rumen microbiota—to understand their roles in marbling—have not been systematically performed. The rumen microbiota of 14 Korean beef cattle (Hanwoo), which showed similar carcass characteristics and blood metabolites but different marbling scores, were analyzed by 16S rRNA gene sequencing. The rumen samples were grouped into two extreme marbling score groups of host animals as follows: LMS, marbling score≤ 4 or HMS, marbling score ≥7. Species richness tended to be higher in the HMS group, whereas the overall microbiota differed between LMS and HMS groups. RFP12, Verrucomicrobia, *Oscillospira*, Porphyromonadaceae, and *Paludibacter* were differentially abundant in the HMS group, whereas *Olsenella* was abundant in the LMS group. Some marbling-associated bacterial taxa also contributed to the enrichment of two lipid metabolic pathways including “alpha-linolenic acid metabolism” and “fatty acid biosynthesis” in the HMS microbiome. Taxonomic drivers of fatty acid biosynthesis, particularly in the rumen microbiome of high-marbled meat, could thus be further studied to increase the intramuscular fat content.

## 1. Introduction

In the beef industry, particularly in Korea and Japan, producers have attempted to increase intramuscular fat. Hanwoo is a breed native to South Korea that has been genetically selected to have a high marbling content. The longissimus muscles of Hanwoo cattle contain 48% oleic acid (% of total fatty acids) [1,2], which is positively associated with meat flavor [3]. Intramuscular fat, known as “marbling,” is highly correlated with meat quality traits (e.g., flavor, tenderness, and juiciness); thus, producers require higher marbling scores to achieve greater economic benefits. By using genetic approaches to detect single-nucleotide polymorphisms potentially related to meat quality [4,5,6], many researchers have attempted to increase marbling in beef. Further to this, various dietary oil supplements have been widely used to reduce levels of saturated fatty acids in adipose tissues [7,8,9,10,11]. Other dietary interventions have also been introduced, such as excluding vitamin A supplementation [12,13] and high-grain diet feeding, particularly at early ages [14,15,16,17,18]. However, few studies have evaluated the rumen microbiota and its relationship with marbling or meat quality.

Early research demonstrated that marbling adipose tissue favors the use of glucose for fat synthesis, especially at early age [19]. During later phases of growth, the marbling adipose tissue also uses acetate as a secondary carbon source, which contributes less than 20% to adipogenesis [19]. Because the ruminal fatty acid composition affects the availability of carbon sources for adipose tissues, including both intramuscular and subcutaneous fat [20], shifts in rumen microbiota might be associated with lipogenic metabolism in these tissues.

Biohydrogenating bacteria are the most widely studied rumen microbiota with respect to fatty acid metabolism in the rumen. Rumen biohydrogenating bacteria represented by *Butyrivibrio* spp. [21,22], *Megasphaera elsdenii* [23], and *Propionibacterium* spp. [24] detoxify dietary polyunsaturated fatty acids. These bacterial species provide a variety of biohydrogenation intermediates including *trans*-18:1, conjugated linoleic acid, and conjugated linolenic acid [25], which can be delivered to the lower gut and affect adipocyte differentiation either positively or negatively [26].

Many studies have evaluated correlations between the rumen microbiome, mostly based on the taxonomic abundances of identified microbiota, and production factors such as the feed efficiency of beef cattle [27]. Moreover, previous reports identified some rumen microbial taxa associated with fatty acid composition in the subcutaneous fat of steers fed different diets [28,29]. These results demonstrated the potential effect of the rumen microbiome on fatty acid metabolism in adipose tissue; however, correlations between intramuscular fat differentiation or marbling score and microbial factors remain unclear.

We hypothesized that rumen microbial populations affect the marbling of Korean beef cattle, either positively or negatively, based on correlations between microbial abundance and carcass characteristics associated with marbling scores among 14 slaughtered Korean beef cattle. We further predicted which microbial factors might be responsible for the high and low marbling of the meat, particularly determined based on the marbling score. The metabolites and predicted functional pathways used, based on differentially abundant microbial taxa, were considered candidates that affect marbling. This study was based on previously published experiments demonstrating the effect of early weaning dietary regimens on adipogenic gene expression profiles during the fattening stage of Hanwoo cattle [30].

## 2. Materials and Methods

### 2.1. Animals and Diet

All experimental procedures were approved and conducted under the guidelines of the National Institute of Animal Science Institutional Animal Use and Care Committee in Korea (approval no.: 2016-205). The 25 Hanwoo steers included were slaughtered before morning feeding at 28 months of age, with an average weight of 626.08 kg. Before slaughter, blood samples were collected from the 25 steers via jugular venipuncture into a Vacutainer to analyze the blood biochemistry parameters. Glucose, triglyceride, cholesterol, high-density lipoprotein, and low-density lipoprotein were analyzed using a Cobas 8000 c702 analyzer (Roche Diagnostics, Mannheim, Germany). Insulin was determined using a VersaMax Microplate Reader (Molecular device, Sunnyvale, CA, USA), whereas total lipid was analyzed using the Agilent 8453 spectrophotometer (Agilent, Waldbronn, Germany). Carcass characteristics were measured and the marbling score was evaluated in accordance with the Beef Marbling Standard (BMS) of South Korea [31]. Briefly, quality grade (QG) was used to assign cattle to five groups as follows: (1) QG 1++ (BMS 8 or 9), (2) QG 1+ (BMS 6 or 7), (3) QG 1 (BMS 4 or 5), (4) QG 2 (BMS 2 or 3), and (5) QG 1 (BMS 1). Fourteen of 25 steers were selected and divided into two groups based on their marbling score, including low-marbling score (LMS, marbling score ≤4) and high-marbling score (HMS, marbling score ≥7) cohorts, to characterize ruminal microbiota in accordance with the two extreme marbling scores.

The Hanwoo steers used in this study were fed four different diets for 10 weeks during the early weaning period. Thereafter, all cattle were fed the same diet primarily comprising corn, wheat bran, and corn-gluten feed. Water and total mixed rations were provided ad libitum. Appendix A shows the feed compositions used in the early weaning period and in the fattening stages. The early weaning diets did not significantly alter the carcass characteristics, blood metabolites, and overall microbiota in both groups (Table 1 and Appendix A).

### 2.2. DNA Extraction from Rumen Samples

The rumen samples for DNA extraction were collected directly after slaughter. The RBB+C bead-beating method [32] was used to extract metagenomic DNA from the rumen samples. The quality and quantity of extracted DNA were analyzed by agarose gel (1%) electrophoresis and with a NanoDrop ND-2000 Spectrophotometer (NanoDrop Technologies, Wilmington, DE, USA).

### 2.3. 16S rRNA Gene Amplicon Sequencing

Amplicons targeting the V3-V4 region of 16S rRNA genes were prepared and sequenced (Macrogen, Daejeon, Korea). Briefly, indexed 16S rRNA gene amplicon libraries from both bacteria and archaea were amplified using 341F (5’-CCTACGGGNGGCWGCAG-3’) and 805R (5’-GACTACHVGGGTATCTAATCC-3’) universal primers [33] with a unique barcode for each DNA sample. The amplicon libraries were sequenced using the 2 × 300 paired-end protocol on the Illumina MiSeq platform (San Diego, CA, USA). After demultiplexing the sequenced amplicons, the barcode and primer sequences were removed using Cutadapt [34]. The available plugins within QIIME2 (version 2019.4) [35] were used to analyze the amplicon sequences. First, the DADA2 plugin was used to denoise the forward and reverse reads with quality filtering (Q > 25) and merged, which was followed by chimera removal [36]. Raw amplicon sequence data are available from the NCBI sequence read archive under the accession number PRJNA523867.

### 2.4. Taxonomic and Diversity Analyses

The amplicon sequencing variants (ASVs) were assigned to the pre-trained Greengenes (13_8 version) reference database, which was trimmed in the V3-V4 primer regions using the naïve Bayesian taxonomic classifier [37]. ASVs that matched mitochondria, chloroplasts and unassigned variants of a known taxon were excluded from downstream analysis. Major classified phyla, families, and genera present in >50% of the 14 rumen samples were mainly evaluated in this study. The alpha-diversity of each sample was examined for species richness, evenness, phylogenetic diversity, Shannon’s index, and Simpson’s index based on rarefied ASV tables using 78,006 sequences per sample. Principal coordinates analysis (PCoA) was computed based on both the unweighted and weighted UniFrac distance matrices to compare the dissimilarity of prokaryotic microbiota among different marbling score groups. The linear discriminant analysis effect size (LEfSe) method was used to identify differentially abundant taxa between LMS and HMS groups with a linear discriminant analysis score of 2 as the cutoff [38]. The number of shared and exclusively identified taxa between LMS and HMS groups at the level of the collapsed phylum, family, and genus was determined and visualized using a Venn diagram.

### 2.5. PICRUSt2 Functional Prediction

ASVs and the BIOM table of the ASVs were used to predict functional features from 16S rRNA gene data using PICRUSt2 [39] with the default pipeline. Briefly, input ASVs were placed into reference sequence alignment, which was followed by submitting these sequences to a reference phylogeny to build a tree file. Next, gene family abundances were predicted for each ASV using the generated tree file as input with pre-calculated 16S- and KEGG ortholog- and Enzyme Commissions (EC)-count tables as references. Gene family copy abundances were normalized to the 16S copy number of each ASV, and KEGG ortholog and EC profiles for each sample were predicted using the ASV abundance BIOM table. Principal components analysis (PCA) was performed to analyze the overall dissimilarity of predicted KEGG orthologs and EC among different marbling score groups.

### 2.6. Identification of Taxonomic Drivers of PICRUSt2-Predicted Functions Using FishTaco

To identify individual functional shifts between LMS and HMS groups and their taxonomic contributors, Functional Shifts’ Taxonomic Contributors (FishTaco) was used [40]. Using PICRUSt2 to predict KEGG orthologs as the functional abundance profile with taxonomic profiles at the genus level, FishTaco links these inputs by inferring genomic content de novo. The taxonomic abundance of major classified genera present in at least 50% of rumen samples was used. Lipid metabolism enriched in the HMS group was visualized with taxonomic contributions using the FishTacoPlot R package (https://github.com/borenstein-lab/fishtaco-plot).

### 2.7. Statistical Analyses

Measures of alpha diversity, meat quality evaluations, carcass characteristics, and blood metabolites were analyzed using the GLIMMIX procedure in SAS 9.4 (SAS Institute, Inc., Cary, NC, USA). Comparisons were made between low- and high-marbling score groups (LMS and HMS). The Tukey post-hoc test was used to find significant differences between the measurements for each comparison. Beta-diversity analysis was performed via permutational multivariate analysis of variance (PERMANOVA) implemented in QIIME2 with both unweighted and weighted UniFrac distance matrices. PERMANOVA was used to perform the multivariate statistical analysis of functional dissimilarity based on Bray–Curtis as a similarity index with 9999 permutations using PAST3 software [41]. Pearson correlation coefficients between meta quality evaluations and the relative abundance of major classified microbial genera were computed using the PROC CORR procedure in SAS and visualized using the R (v. 3.5.0) package “corrplot”. Significant differences were considered at *p* ≤ 0.05 with a tendency for significance indicated by 0.05 < *p* ≤ 0.1.

## 3. Results

### 3.1. Alpha- and Beta-Diversity Analyses

Microbiota was examined and compared in terms of both alpha and beta diversity among the two marbling score groups. All quality- and taxonomy-filtration procedures resulted in at least 78,006 sequences from all rumen samples (Appendix A). Diversity analysis was performed with rarefied BIOM tables using the lowest sequencing depth among the samples for rarefaction to normalize the ASVs per sample. More than 99.9% Good’s coverage was observed for all samples (data not shown). Both observed ASVs and Chao1 estimates tended to be higher in the HMS group than in the LMS group (Table 2). PCoA based on the unweighted UniFrac distance matrices showed different overall microbiota between the LMS and HMS groups, but computing these values based on taxa abundances offset the significance (Figure 1).

### 3.2. Taxonomic Analysis with Major Classified Taxa

The Venn diagram revealed numerous taxa associated with each marbling score group (Figure 2). At the phylum level, 21 of 23 phyla were shared among the two marbling score groups. Further to this, 87 families and 143 genera were shared by the two marbling score groups. Taxa in specific marbling score groups were observed more at lower taxa levels. There were 12 LMS-specific families corresponding to 29 genera, whereas 14 families corresponding to 27 genera were HMS-specific. The taxonomic abundances of major classified taxa at the phylum, family, and genus levels with respect to the two marbling score groups can be visualized by the provided barplots (Appendix A). Among the major classified taxa, RFP12, Verrucomicrobia, *Oscillospira*, Porphyromonadaceae, and *Paludibacter* were differentially abundant in the HMS group, whereas the abundance of *Olsenella* was enriched in the LMS group (Figure 3). No archaeal taxa differed based on marbling score groups in this study.

### 3.3. Functional Prediction and Taxonomic Contributors to Lipid Metabolism

The overall distribution of functional features based on the predicted KEGG orthologs and EC from both representative sequences and the taxonomic abundance of each taxon can be visualized in PCA plots (Figure 4). Based on PERMANOVA tests, no clear difference in overall function was observed between LMS and HMS groups (Figure 4). Using FishTaco comparative analysis between LMS and HMS groups to identify both differentially abundant taxonomic and functional features, two functional pathways of lipid metabolism, namely “alpha-linolenic acid metabolism” and “fatty acid biosynthesis,” were enriched based on the major genera in the HMS group (i.e., marbling score ≥7; Figure 5). Those two pathways related to lipid metabolism were concurrently enriched by *Treponema* as the major taxonomic driver in the HMS group. Moreover, *Oscillospira* specifically contributed to the enrichment of fatty acid biosynthesis in the HMS group. In contrast, higher abundances of *Succiniclasticum*, *Moryella*, *Butyrivibrio*, and *Bulleidia* in the LMS group inversely contributed to the enrichment of these pathways in the HMS group due to their relatively limited contribution to these functions. There were no enriched lipid metabolic pathways in the LMS group (data not shown).

### 3.4. Correlation between Major Classified Genera and Meat Quality Measurements

Significant correlation coefficients among backfat thickness, marbling scores, and major classified genera across all rumen samples are shown in Figure 6. *Streptococcus* and *Ruminobacter* were negatively associated with backfat thickness, whereas candidate genus SHD-231 showed a positive association. *Paludibacter* and L7A_E1 were positively associated with the marbling score and *Succiniclasticum*, *Bulleidia*, and *Sutterella* showed negative associations.

## 4. Discussion

Increasing meat quality is a goal of the animal industry to increase market value. Meat quality is typically defined by the marbling score, which is equivalent to the amount of intramuscular fat. Studies on the impact of biohydrogenating bacteria on intramuscular fat deposition have provided controversial results [26], and few studies have examined the relationship between the rumen microbiome and marbling score, particularly in beef. The rumen is the primary location where feed ingredients are digested mainly into volatile fatty acids (VFAs), which are further converted into glucose and fatty acids [42]. The animals in this study were fed the same diet for 22 months after the first 10 weeks of weaning, and no differences in meat quality indices were observed except for marbling between the LMS and HMS groups. Furthermore, no significant effect of different early weaning diets was observed on marbling or other meat-quality properties, as well as on overall rumen microbiota. Therefore, we predicted that differences in marbling score were correlated with the rumen microbiome.

The rumens of animals producing meat with higher marbling tended to have more diverse prokaryotic microbiota, based on richness indices, than the rumens of cattle producing less marbled meat. A more diverse microbiota was previously correlated with feed inefficiency, resulting in significantly more complex metabolites produced by various functional pathways [43,44]. Previous studies also suggested an association between the marbling score and an inefficient microbiome [45,46,47]. However, other studies observed no differences in marbling scores across different feed efficiency groups of beef cattle [48,49].

Among the differentially abundant taxa listed for the LMS group, *Olsenella* is considered a lactate and succinate producer [50] and its abundance continuously increases during fattening in high-grain diet-fed beef cattle [51]. HMS-enriched *Paludibacter*, a bacterial genus of Porphyromonadaceae, ferments glucose to produce acetate and propionate as the major fermentation products [52], the former of which is a lipogenic precursor. Particularly, RFP12, which represents >80% of the abundance of Verrucomicrobia, was previously designated core-heritable microbiota in dairy cows corresponding to levels of methane and rumen and blood metabolites, as well as milk production [53]. Because host specificity strongly affects an individual’s microbiota [54], the selection of Hanwoo beef based on the composition of the rumen microbiota might increase marbling in meat at the fattening stage.

Among the rumens of 14 beef cattle, some bacterial taxa were exclusively found in each marbling score group. These unique taxa resulted in a significantly different phylogenetic distribution between the microbiota of LMS and HMS groups. However, because of the limited abundance of these exclusively identified taxa, limited sequencing depth might explain the absence of these minor taxa in the other marbling score group. Although these taxa are not considered major ruminal taxa (<0.01%), they might be candidate contributors to fatty acid metabolism during fat deposition. In addition to the minor groups, more than 49% of genera were unclassified in an average of 14 rumen samples, which might affect various functions related to host production. However, among all unclassified taxa, only two taxa (unclassified [Paraprevotellaceae] and unclassified Mollicutes) were enriched in either LMS or HMS, and these have not been well characterized in the rumen.

The significant correlation between rumen microbial taxa and marbling score revealed the potential roles of these bacteria in controlling intramuscular fat deposition. The relative abundance of *Bulleidia* and L7A_E11 affected the marbling score differently. These genera are phylogenetically located within the Erysipelotrichaceae family, which was recently expanded based on genome recovery from the metagenome of the cow rumen [55]. This family is thought to be associated with disease and related to protein and energy metabolism in the animal microbiome [56,57]. *Bulleidia* was previously predicted to play a biohydrogenating role in that its relative abundance is positively correlated with biohydrogenation intermediates including *trans*-10, *cis*-12 conjugated linoleic acid in dairy cows [58]. The relative abundance of the candidate genus L7A_E11 was 4.21-fold higher in the HMS group, but this requires further evaluation to select this genus as a potential biomarker of marbling. Species of *Butyrivibrio* and *Clostridium* contribute to ruminal biohydrogenation [59]. Limiting the biohydrogenation of unsaturated fatty acids by lowering the abundance of these genera in the HMS group would provide a higher unsaturated fatty acid supply to the adipose tissue, which could drive adipose cell differentiation.

By analyzing both taxonomic abundance and functional enrichment systematically [40], FishTaco identified microbial taxa driving or attenuating specific functional pathways defined in the HMS microbiome. *Treponema* and *Oscillospira* contribute to the two types of lipid metabolism. Some species of *Treponema* require long-chain fatty acids for their growth [52]. The cecal abundance of *Oscillospira* increases during the latter fattening stages of geese [60], whereas its ruminal abundance is maximized in sheep when they graze on green pasture [61]. The abundance of this genus was strongly correlated with the concentration of hexadecenoic acid and oleic acid in the rumens of long-term high-concentrate-fed goats [62]. Other taxonomic drivers might act either as biohydrogenating or propionate-producing bacteria to support the differentiation of marbling adipose tissue. Both taxonomic drivers and reducers of lipid metabolism are putative targets to improve marbling. The relative abundances of several taxonomic drivers were lower in HMS-microbiota but positively contributed to the enrichment of lipid metabolism, including alpha-linolenic acid metabolism and fatty acid biosynthesis in the HMS-microbiome. Among them, *Butyrivibrio* sp., and particularly *Butyrivibrio proteoclasticus*, participate in the conversion of vaccenic acid to stearic acid, which limits the supply of monounsaturated fatty acid to the adipose tissue [63]. The lack of a contribution to the fatty acid synthesis pathway by these bacteria inversely contributed to functional enrichment by decreasing their abundance in the HMS group. *Succiniclasticum, Moryella*, and *Bulleidia* also inversely contributed to lipid metabolism in the HMS group, including both fatty acid biosynthesis and alpha-linolenic acid metabolism, by limiting those functions in their genomes.

## 5. Conclusions

Meat production is achieved by complex metabolic pathways, and the metabolites produced by the rumen microbiota possibly affect the marbling score. By analyzing the rumen microbiota in groups distributed based on the two extreme marbling scores, we predicted the potential contribution of the microbiota to the marbling of Hanwoo beef cattle. Few studies have examined the impact of the rumen microbiota on the marbling of beef, and differentially abundant taxa observed between low- and high-marbled meat groups require further analysis. In addition, taxonomic drivers of fatty acid biosynthesis, particularly in the rumen microbiomes of animals producing high-marbled meat, can be targeted to increase the intramuscular fat content. However, large-scale animal trials with additional observations including VFA, lipogenic gene expression profiles and host genetic differences are needed to verify these microbial markers of marbling and overall meat quality in beef cattle.

## Figures and Tables

**Figure 1 animals-10-00712-f001:**
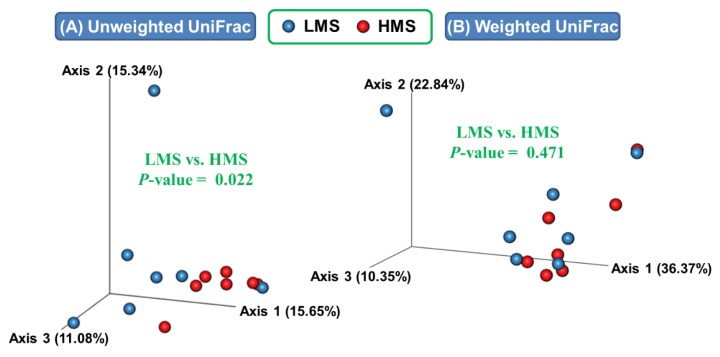
Principal coordinates analysis based on both unweighted (**A**) and weighted (**B**) UniFrac distances of prokaryotic populations in the rumen of Korean beef cattle. PERMANOVA was computed for non-parametric statistical tests between low marbling score (LMS) and high marbling score (HMS) groups.

**Figure 2 animals-10-00712-f002:**
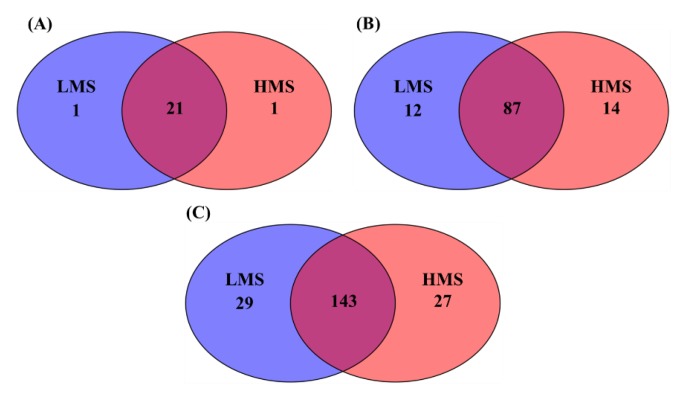
The number of shared and exclusively identified taxa in collapsed (**A**) phyla, (**B**) families, and (**C**) genera BIOM tables between low marbling score (LMS) and high marbling score (HMS) groups.

**Figure 3 animals-10-00712-f003:**
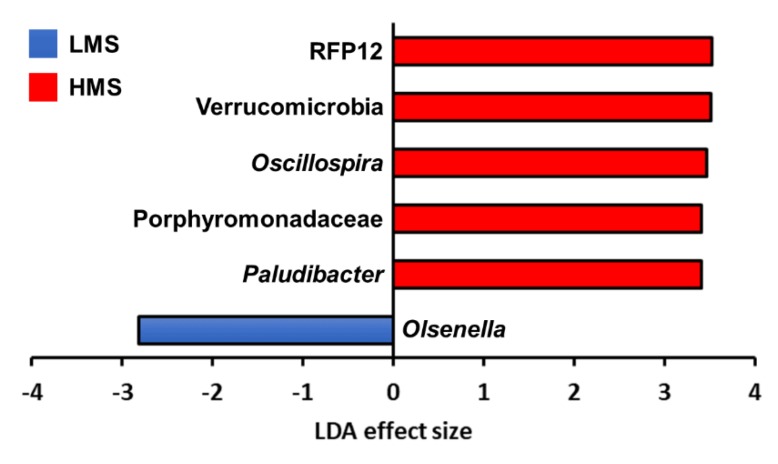
Linear discriminant analysis effect size (LEfSe; differentially abundant bacterial taxa defined by LEfSe analysis) computed between low marbling score (LMS) and high marbling score (HMS) groups.

**Figure 4 animals-10-00712-f004:**
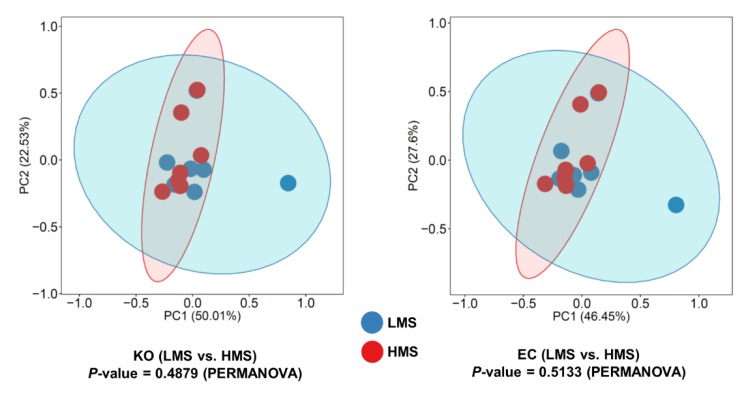
Principal components analysis of PICRUSt2 functional prediction retrieved from prokaryotic populations in the rumens of Korean beef cattle. PERMANOVA was computed for non-parametric statistical tests between low marbling score (LMS) and high marbling score (HMS) groups.

**Figure 5 animals-10-00712-f005:**
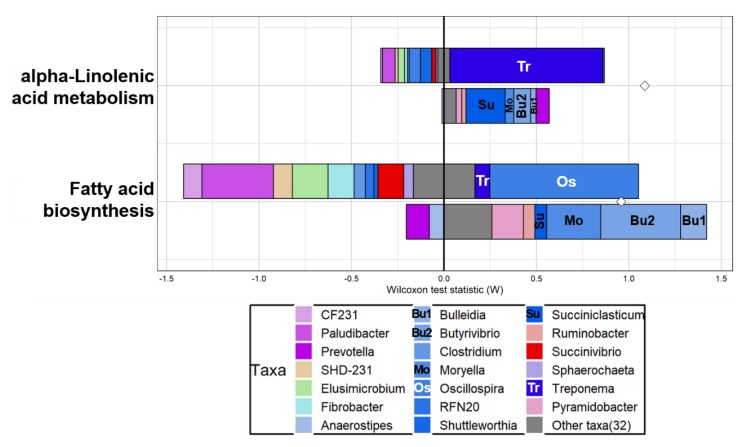
Characterization of taxonomic drivers of two functional pathways related to lipid metabolism in the rumen of the high marbling score (HMS) group of Korean beef cattle using FishTaco.

**Figure 6 animals-10-00712-f006:**
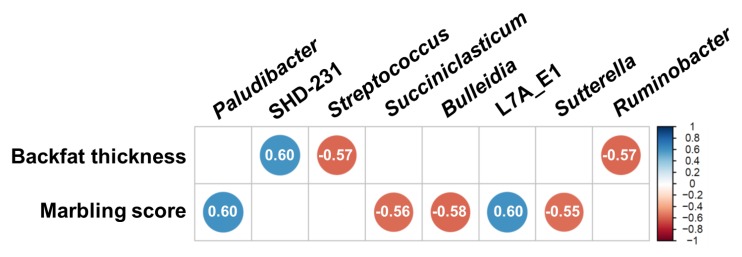
Correlation coefficient matrix between meat quality measurements and relative abundances of major bacterial genera each found in more than 50% of the 14 rumen samples. Only significant correlations (*p* < 0.05, |*r*| > 0.5) are shown. The degree of correlation coefficients are shown using both the color and size of the circles based on the color key on the right side.

**Table 1 animals-10-00712-t001:** Carcass characteristics and blood metabolites of Hanwoo beef cattle between LMS and HMS groups.

Evaluations	Marbling Score Groups		SEM ^3^	*p*-Values		
	LMS ^1^	HMS ^2^		MS	Diet *	MS × Diet
Live weight (kg)	600.29	665.86	17.26	0.2066	0.9777	0.8048
Carcass weight (kg)	331.12	375.76	11.53	0.2702	0.9761	0.9299
Carcass percent (%)	55.16	56.23	0.52	0.8267	0.9032	0.3424
Lean meat weight (kg)	233.50	259.92	8.03	0.3703	0.9746	0.8277
Fat weight (kg)	55.99	68.94	3.13	0.1851	0.9441	0.8723
Bone weight (kg)	43.74	47.79	1.14	0.2617	0.9773	0.9273
Backfat thickness (mm)	10.29	9.57	0.64	0.8693	0.6736	0.9717
Sirloin area (cm^2^)	90.71	100.14	2.90	0.2255	0.8071	0.2504
Meat yield index	71.69	69.44	1.34	0.5246	0.7485	0.5262
Meat color	4.57	4.29	0.14	0.2969	0.1826	0.6559
Fat color	3.14	3.29	0.11	0.5188	0.6885	0.4281
Marbling score **	3.14 ^B^	7.43 ^A^	0.62	<0.0001	0.3742	0.0618
Sirloin crude fat (%)	28.84	35.07	1.69	0.2585	0.9403	0.8442
Renal fat (%)	16.22	14.26	1.01	0.4523	0.9268	0.8684
Cholesterol (mg/dL)	120.17	127.57	8.94	0.3508	0.0884	0.2079
Triglyceride (mg/dL)	22.83	21.57	2.11	0.9478	0.9257	0.6944
High-density lipoprotein (mg/dL)	87.00	106.86	5.70	0.1908	0.2682	0.3136
Low-density lipoprotein (mg/dL)	30.50	33.00	2.79	0.4966	0.2349	0.5109
Total Lipid (mg/dL)	293.83	306.43	20.75	0.3753	0.0826	0.4457
Glucose (mg/dL)	96.67	112.57	7.50	0.7138	0.4471	0.9402
Insulin (μU/ml)	13.27	6.77	2.22	0.2038	0.8244	0.3795

^1^ Low marbling score. ^2^ High marbling score. ^3^ Standard error of the mean. * Diet effect was considered based on the four regimens used for feeding during the early weaning period. Control, mother milk + roughage; T1, milk replacement + concentrate; T2, milk replacement + concentrate + roughage; T3, milk replacement + concentrate + 30% starch. ** Determined based on the beef marling standard [31].

**Table 2 animals-10-00712-t002:** Alpha-Diversity measurements of the prokaryotic microbiota in Hanwoo beef cattle.

Measurements	Marbling Score Groups		SEM ^3^	*p*-Value
	LMS ^1^	HMS ^2^		
Observed ASVs	855.57 ^b^	1039.14 ^a^	49.74	0.0614
Chao1 estimates	860.06 ^b^	1045.96 ^a^	50.39	0.0615
Evenness	0.80	0.82	0.01	0.1959
Faith’s Phylogenetic diversity	64.80	62.14	3.05	0.6818
Shannon’s index	7.76	8.24	0.15	0.1070
Simpson’s index	0.984	0.989	0.002	0.2979

^1^ Low marbling score. ^2^ High marbling score. ^3^ Standard error of the mean. Good’s coverage was >99.9% in all samples. Statistical difference was declared at *p* < 0.05 (uppercase superscript), 0.05 < *p* < 0.1 (lowercase). ASV, amplicon sequencing variant.

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
