# Peer review of "Association between Rumen Microbiota and Marbling Score in Korean Native Beef Cattle"

_animals, 2020, doi:10.3390/ani10040712_

Round 1
Reviewer 1 Report
Title:
Association between rumen microbiota and meat quality in Korean native beef cattle.
This paper investigates the possible association between rumen microbiota and meat marbling score in Hanwoo Korean beef cattle. This paper is well written and deepens the knowledge on the influence of rumen microbiota on ruminant production traits, that has been a hot topic of animal science research in the few last years.
In my opinion this paper deserves to be published, even though there are some minor revisions to be made.
Comments:
L52- 373: Please replace reference number 19 with a more suitable peer reviewed reference
L237: in the discussion there’s no need to refer again to tables and figures, but it is necessary to explain the outputs that have been already described in the “Results” session. Please remove the references to tables and figures here and elsewhere in the “Discussion” session.
In all the figures (1-6) the resolution must be improved.
Author Response
Response to Reviewer 1 comments
Comments:
L52- 373: Please replace reference number 19 with a more suitable peer reviewed reference
Response: Thanks to your suggestion. We replaced the reference to ‘Smith SB and Crouse JD 1984. Relative contributions of acetate, lactate and glucose to lipogenesis in bovine intramuscular and subcutaneous adipose tissue. The Journal of nutrition 114, 792-800.’ (line 54).
L237: in the discussion there’s no need to refer again to tables and figures, but it is necessary to explain the outputs that have been already described in the “Results” session. Please remove the references to tables and figures here and elsewhere in the “Discussion” session.
Response: Now, all the signs for both the ‘Table’ and ‘Figure’ in discussion section were removed.
In all the figures (1-6) the resolution must be improved.
Response: We tried to improve the resolution of all the images. On our side, PDF conversion was OK to maintain the quality of image. If blurred image still happens, we will try to find another way. Please look over the new ones included in the manuscript.
Reviewer 2 Report
I think this study is interesting in microbiota differed LMQ and HMQ.
Can you control the microbiota in the rumen for putting the marbling to the maximum ?
Can you improve the marbling by modifying rumen microbiota in low genetic ability cattle ?
Is there confounding between genetic ability and rumen microbiota for marbling in this study ?
I think you should measure ruminal condition at the same time because you only observe microbiota and suppose intermediate pathway to intramuscular fat deposition.
My specific comments are as follows,
L305; However, large-scale animal trials are needed to verify these microbial markers of marbling and overall meat quality in beef cattle. --- I also think if this observation is reproducible or not.
Figure 6; Why do you not state r in figures ?
Figure S1; I feel there are no differences. Is this all right ?
L325, 327; Korean Journal for Food Science of Animal Resources --- Korean J. Food Sci. Anim. Resour.
L373; Smith, S.; Johnson, B. --- Smith, S.B.; Johnson, B.J.
L374; Centennial, CO, USA: National Cattlemen’s Beef Association; 2014. ?
L379; The Journal of Nutrition --- J. Nutr.
L393; Journal of Animal Science and Biotechnology --- J. Anim. Sci. Biotechnol.
L409; Animal Frontiers --- Anim. Front.
L413; The ISME journal --- ISME J.
L429; BioRxiv --- bioRxiv ?
L430; 10.1101/672295, 672295, doi:10.1101/672295. --- doi:10.1101/672295. ?
L432; Cell Host & Microbe --- Cell Host Microbe
L434; Palaeontologia Electronica --- Palaeontol. Electron.
L436; Animal Frontiers --- Anim. Front.
L439; The ISME journal --- ISME J.
L443; PLoS Genet --- PLoS Genet.
L452; The Professional Animal Scientist --- Prof. Anim. Sci.
L459; Journal of Animal Science and Biotechnology --- J. Anim. Sci. Biotechnol.
L472; Science advances --- Science Advances
L478; Nature communications --- Nature Communications
L479; Frontiers in Cellular and Infection Microbiology --- Front. Cell. Infect. Microbiol.
L488; Antonie Leeuwenhoek --- Antonie van Leeuwenhoek
Author Response
Reviewer 2
Comments and Suggestions for Authors
I think this study is interesting in microbiota differed LMQ and HMQ.
Can you control the microbiota in the rumen for putting the marbling to the maximum ?
Can you improve the marbling by modifying rumen microbiota in low genetic ability cattle ?
Is there confounding between genetic ability and rumen microbiota for marbling in this study ?
I think you should measure ruminal condition at the same time because you only observe microbiota and suppose intermediate pathway to intramuscular fat deposition.
Response to Reviewer 1 comments
Response: As noted in response to the other reviewer, we no longer have these samples therefore unfortunately, going back to look at VFA proportion is not possible. We agree that knowing the VFA proportion was useful to understand what happens in the rumen of two different meat-quality groups. The present study may be used as a basis for further experiments using similar methods with more observations including VFA and other measurements to determine ruminal conditions.
My specific comments are as follows,
L305; However, large-scale animal trials are needed to verify these microbial markers of marbling and overall meat quality in beef cattle. --- I also think if this observation is reproducible or not.
Response: This is correct. That is why we suggested the large-scale experiment in conclusion. We also think more supporting observations (e.g., VFA profiles, lipogenic gene expressions, and host genetic differences, etc) will provide better insights to determine the contribution of microbial factors on the intramuscular fat depositions and marbling.
Figure 6; Why do you not state r in figures ?
RESPONSE: Now the explanation about the color key on the RHS and correlation coefficients were added (line 235).
Figure S1; I feel there are no differences. Is this all right ?
Response: The barplots were included to help readers see the relative abundances of major microbial taxa. Because of the nature of relative abundances, especially in the barplots, the majority of the spaces were given to few taxa so that it is not easy to distinguish the differences of minor taxa in between two groups. As the reviewer felt that we didn’t see many differentially abundant taxa but the statistically different taxa were included in Figure 3 supported by LEfSe analysis.
L325, 327; Korean Journal for Food Science of Animal Resources --- Korean J. Food Sci. Anim. Resour.
Response: Revised as suggested (line 344 & lines 345 - 346).
L373; Smith, S.; Johnson, B. --- Smith, S.B.; Johnson, B.J.
Response: Now the reference was removed.
L374; Centennial, CO, USA: National Cattlemen’s Beef Association; 2014. ?
Response: Now the reference was removed.
L379; The Journal of Nutrition --- J. Nutr.
Response: Revised as suggested (line 393, 395).
L393; Journal of Animal Science and Biotechnology --- J. Anim. Sci. Biotechnol.
Response: Revised as suggested (line 409).
L409; Animal Frontiers --- Anim. Front.
Response: Revised as suggested (line 425).
L413; The ISME journal --- ISME J.
Response: Revised as suggested (line 429).
L429; BioRxiv --- bioRxiv ?
Response: Revised as suggested (line 445).
L430; 10.1101/672295, 672295, doi:10.1101/672295. --- doi:10.1101/672295. ?
Response: Revised as suggested (line 446).
L432; Cell Host & Microbe --- Cell Host Microbe
Response: Revised as suggested (line 448).
L434; Palaeontologia Electronica --- Palaeontol. Electron.
Response: Revised as suggested (line 450).
L436; Animal Frontiers --- Anim. Front.
Response: Revised as suggested (lines 451 - 452).
L439; The ISME journal --- ISME J.
Response: Revised as suggested (line 455).
L443; PLoS Genet --- PLoS Genet.
Response: Revised as suggested (line 459).
L452; The Professional Animal Scientist --- Prof. Anim. Sci.
Response: Revised as suggested (line 468).
L459; Journal of Animal Science and Biotechnology --- J. Anim. Sci. Biotechnol.
Response: Revised as suggested (line 474 - 475).
L472; Science advances --- Science Advances
Response: Revised as suggested (line 489).
L478; Nature communications --- Nature Communications
Response: Revised as suggested (line 495).
L479; Frontiers in Cellular and Infection Microbiology --- Front. Cell. Infect. Microbiol.
Response: Revised as suggested (line 496).
L488; Antonie Leeuwenhoek --- Antonie van Leeuwenhoek
Response: Revised as suggested (line 505).
Response: All the information on references were corrected as suggested
Reviewer 3 Report
Overall, text should be revised in an extensive way and below are some of the specific comments:
1.Please provide the methods for the evaluation of blood metabolites and marbling grading standards.
2. L101:The rumen samples were collected directly after slaughter. Why not collecting rumen fluid and blood after feeding?
3. Please provide the data of volatile fatty acids.
4. The discussion section needs to be substantially revised. The chosen references are not adequate for the topic of the paper. Please summarize the main contribution of each referenced paper. Please show how it is different from the others and why it deserves mentioning.
5. RFP12, Verrucomicrobia, Oscillospira, Porphyromonadaceae, and Paludibacter were differentially abundant in HMQ, and Olsenella was abundant in LMQ. Please provide possible reasons that these bacteria promote or depress intramuscular fat deposition.
Author Response
Reviewer 3
Comments and Suggestions for Authors
Overall, text should be revised in an extensive way and below are some of the specific comments:
Response to Reviewer 3 comments
1.Please provide the methods for the evaluation of blood metabolites and marbling grading standards.
Response: Revised as suggested (lines 86 - 91).
- L101:The rumen samples were collected directly after slaughter. Why not collecting rumen fluid and blood after feeding?
Response: In the present study, cannulated cattle were not used. The Rumen Microbial Genomics Network indicated that collection of rumen contents from slaughtered cattle can be an alternative method for microbial studies if rumen sampling via cannulation is not possible. Although the stomach tubing method could be used for rumen sampling in the present study, we did not use the stomach tubing to avoid a potential contamination of rumen samples with saliva. We think that it is difficult to tell which rumen sampling method is “right” or “wrong”. Blood samples were collected before slaughter for analysis of blood chemistry parameters. We revised the text (lines 86 - 91).
- Please provide the data of volatile fatty acids.
Response: As noted in response to the other reviewer, we no longer have these samples therefore unfortunately, going back to look at VFA proportion is not possible. We agree that knowing the VFA proportion was useful to understand what happens in the rumen of two different meat-quality groups. The present study may be used as a basis for further experiments using similar methods with more observations including VFA and other measurements to determine ruminal conditions.
- The discussion section needs to be substantially revised. The chosen references are not adequate for the topic of the paper. Please summarize the main contribution of each referenced paper. Please show how it is different from the others and why it deserves mentioning.
Response: Now we revised the manuscript and discussion to focus on the association between microbiota and marbling not with the overall meat quality.
And we have reviewed the references and revise the discussions. We added more details of some of the cited references only when it needs to. Some of inappropriate references were removed or replaced. If it still needs further revision, please specify the references, lines or sentences.
- RFP12, Verrucomicrobia, Oscillospira, Porphyromonadaceae, and Paludibacter were differentially abundant in HMQ, and Olsenella was abundant in LMQ. Please provide possible reasons that these bacteria promote or depress intramuscular fat deposition.
Response: We have provided possible reasons of their dominance either in LMS or HMS based on the previous studies and those fermentation characteristics. As long as we searched, there is limited information about RFP12 which is a candidate family within Verrucomicrobia. Direct evidence of their contribution to the fat deposition still unclear so that we only could provide their differential abundance in various conditions and, their fermentation characteristics which might be linked to the lipid metabolism and fat deposition.
Reviewer 4 Report
Dear Authors,
attached please find a pdf with comments.
Regards

Author Response
Reviewer 4
Response to Reviewer 4 comments
Highlighted fragments may require revision of grammar/spelling/word order/punctuation
Response: Thanks for your suggestions. We have revised the indicated sentences.
longissimus muscle : which muscle exactly, or is it plural?
Response: This is plural (line 43).
Smith and Johnson (2014)[19] : Incorrectly quoted
Response: Thanks to your suggestion. We replaced the reference to ‘Smith SB and Crouse JD 1984. Relative contributions of acetate, lactate and glucose to lipogenesis in bovine intramuscular and subcutaneous adipose tissue. The Journal of nutrition 114, 792-800.’ (line 54).
corn-gluten feed : is this ok?
Response: Corn-gluten feed is a byproduct of corn syrup production. This is OK.
(https://github.com/borenstein-lab/fishtaco-plot). Wouldn't this link fit better the Reference section?
Response: Since this R package is not published but implemented in FishTaco website, if there is no limitation to add hyperlink in the manuscript, we want to keep it as the current format.
Image blurred - low quality
Response: We tried to improve the resolution of all the images. On our side, PDF conversion was OK to maintain the quality of image. If blurred image still happens, we will try to find another way. Please look over the new ones included in the manuscript.
Line ??: and overall -> and on overall
Response: Revised as suggested (line 251).
Please go through the list of references and correct some of the abbreviations for journals. See https://www.mdpi.com/journal/animals/instructions.
Response: All the reference formats were revised as suggested.
Round 2
Reviewer 3 Report
The author has revised the manuscript according to the my comments.
I don't have more questions except that: in Table 1. "Carcass percent" should be "Dressing percentage" and "Sirloin area" should be "ribeye area'?